

# Injuries among adolescents in Greenland: behavioural and socio-economic correlates among a nationally representative sample

Louis Jansen[1], Till Bärnighausen[1] and Michael Lowery Wilson[1,2]

[1] Heidelberg Institute of Global Health, Ruprecht-Karls-Universität Heidelberg, Heidelberg, Baden-Wuerttemberg, Germany
[2] Injury Epidemiology and Prevention (IEP) Research Group, Turku Brain Injury Centre, Department of Clinical Neurosciences, Turku University Hospital, University of Turku, Turku, Finland

## ABSTRACT

**Background:** Injuries are among the most important threats to adolescent health, making examination of the patterns and risk factors a critical area of research. There exists a paucity of information on the health and injury experience of school-attending adolescents in Greenland. Consenting Greenlandic schoolchildren ($n$ = 2,254) aged 9–19 years were included in the Health Behavior in School-Aged Children study 2005/2006. The aim of this study was to examine the socio-economic and behavioural correlates that were associated with injury occurrence among school-attending Greenlandic adolescents.

**Methods:** This study made use of two multinomial regression models to examine injury occurrence regarding potential influencing factors such as physical activity, risk behaviours, bullying and family socio-economic status (SES).

**Results:** Those self-reporting 1–2 injuries within the recall period were more likely to be male (OR = 1.70; CI [1.39–2.09]), involved in physical fighting (OR = 1.82; CI [1.33–2.47]), bullied (OR = 1.81; CI [1.47–2.24]) and participated in bullying others (OR = 1.53; CI [1.25–1.89]). Those reporting three or more injuries were again mostly male (OR = 2.13; CI [1.44–3.14]), involved in physical fighting at higher rates (OR = 4.47; CI [2.86–7.01]), bullied more often (OR = 2.43; CI [1.65–3.57]) and were more likely to bully others (OR = 1.67; CI [1.13–2.45]). Living without a mother proved to be significantly correlated with suffering 3 or more injuries during the recall period (OR = 1.63; CI [1.05–2.52]). The study results support the idea that factors that were found to be associated with injury occurrence, such as bullying and aggressive behaviour, should be taken into account when conducting future research on the nature of injuries among Greenlandic adolescents. More research on this topic is needed to identify factors that might modify the associations between injuries and adolescent behaviour and SES.

## INTRODUCTION

Injuries among adolescents has emerged as a global problem of public health importance. It is estimated by the World Health Organisation (WHO), that injuries account for almost

Corresponding author
Louis Jansen, louis.jansen@gmx.net

one million deaths annually among young people under the age of 18 years worldwide, making it one of the most urgent adolescent health concerns (*Djerboua, Chen & Davison, 2016*). From the age of one year until adulthood, injuries account for a major portion of adolescent deaths. Even though ways to decrease the likelihood and severity of injuries are known, prevention and awareness still trail behind the current state of research (*Peden et al., 2009*). Considering that effective measures to prevent injuries do exist, it must be a focal point of injury research to determine what puts adolescents at risk for and protects them against injuries.

Injuries can be divided into two major areas: those which are violence-related or intentional injuries and unintentional injuries, so called 'accidents'. As both the World Report on Violence against Children and the World Report on Child Injury Prevention state, prevention of violence and injury respectively is the key for improving adolescents' well-being (*Peden et al., 2009*; *Pinheiro & Children UNS-GsSoVA, 2006*).

The most common causes of unintentional injuries among adolescents are road traffic injuries, drowning, and other causes (including categories such as smothering, asphyxiation, bites, hypo/hyperthermia and natural disasters). Risk factors these causes share are poverty, lack of supervision or care and often but not always male gender (*Peden et al., 2009*). In high-income countries road traffic and drowning related injuries account for more than two thirds of all unintentional injury deaths among people aged less than 20 years. To compare, in low- and middle-income countries other causes (including categories such as smothering, asphyxiation, bites, hypo/hyperthermia and natural disasters) and road traffic injuries account for more than 60% of all deaths in under 20 year olds (*Harvey et al., 2009*).

Interpersonal violence and injuries can lead to harm that threatens the physical and psychological integrity in ways that range from harmless to possibly life-threatening. The devastating effect of adolescent violence was described by *David-Ferdon & Simon (2014)*, when they showed that more U.S. youth die from homicide each year than from cancer, heart disease, birth defects, flu and pneumonia, respiratory diseases, stroke, and diabetes combined. International research has shown that social marginalisation and adverse familiar circumstances (e.g. familiar abuse and neglect) as well as drug abuse and economic inequality can be considered as risk factors for being involved in violence (*David-Ferdon & Simon, 2014*; *Kobusingye et al., 2010*). Protective factors include a safe and care-giving familiar environment, strong social bonds outside of the family, an understanding of responsible drug use, and high socio-economic status (SES) (*Bushman et al., 2016*).

Despite the importance of research on injury and violence among Greenlandic adolescents, research on it remains sparse. The last English-language paper addressing the topic of adolescent aggression, which is one of many risk factors for adolescent injuries (*Laflamme et al., 2002*), was written by *Schnohr & Niclasen (2006)*. The authors reported an increase in the prevalence of bullying in Greenlandic schools. In comparison to 35 other countries (located in Europe and North America) Greenland ranked 7th in terms of bullying others several times during the month preceding survey. Greenlandic schoolchildren showed above-average percentages for bullying others as well (*Schnohr &*

*Niclasen, 2006*). Those numbers stress the need of further research to be conducted on the topic of injuries among Greenlandic adolescents and their socio-economic and behavioural correlates. The aim of this study was to examine the socio-economic and behavioural correlates for injuries among a nationally representative sample of school-attending adolescents in Greenland.

## MATERIALS AND METHODS

### Setting

The current study is based on data collected in Greenland, the world's largest island. Greenland is an autonomous territory within the Kingdom of Denmark. The population is 56,225, making it the most sparsely populated territory on earth. Children and adolescents of school age, classified as being between 5 and 19 years of age, make up a total of 11,291 inhabitants, accounting for 20.1% of the total population (*Blaabjerg, 2019*). The gross domestic product (GDP) per capita is USD$ 41,800 (*Agency & Office, 2016*) and the Human Development Index (HDI) is 0.786 (*Avakov, 2015*), which both classify Greenland as a highly developed country.

### Sample

In Greenland, 2,254 secondary school students (52% females) aged 9–19 years with a mean age of 13.4 years participated in the survey. The final sample included 2,217 participants, as 37 participants had to be excluded since they did not have complete information on the variables age or gender. All Greenlandic schools with more than ten students were invited to take part in the survey. If the school board had decided to participate, they were then provided with the questionnaires. A Greenlandic- and Danish-language paper questionnaire was administered by a teacher. Detailed information about the questionnaire, data collection methods and study design can be found on the Health Behavior in School-Aged Children (HBSC) website—http://www.hbsc.org/ and the HBSC Greenland website—http://www.hbscgreenland.org/. Passive consent from adolescents participating in the HBSC survey was obtained. The resulting dataset was stripped of potentially identifying characteristics. No individual participant was contacted during or after the research period.

### Measurements

The data for this study was derived from Greenland's contribution to the HBSC in 2005/2006. The HBSC study was adopted by the WHO in 1983 to cross-sectionally survey 11, 13 and 15 year-old boys' and girls' health and well-being, social environments and health behaviours. Since then it has been conducted every four. The questions that were analysed in this study are depicted in Appendix 1. The question, the dependent variable injuries was derived from, 'During the past 12 months, how many times have you been injured?' had five response options. Those were '1 = I have not been injured', '2 = Once', '3 = Twice', '4 = 3 times' and '5 = 4 times or more'. Injuries were defined in the official HBSC survey form as injuries that required medical attention. The literal question specifically included injuries that resulted from playing sports, fighting with someone else at different

places such as the street or at home, being poisoned, or being burned. Illnesses such as measles or the flu were specifically excluded in the literal question, while both intentional and unintentional injuries were included. In accordance with suggestions from the peer-reviewed literature, we examined the following independent variables: gender, age, vigorous physical activity (VPA), VPA (outside school hours), smoking, alcohol misuse, physical fighting, bullying victimisation, bullying perpetrator, parental job, living with parents and self-assessed wealth (*Acquah, Wilson & Doku, 2014*; *Acquah et al., 2014*; *Celedonia et al., 2013*; *Garmy, Vilhjalmsson & Kristjansdottir, 2018*). The relevant questions and examined variables and coding can be found in the appendix under the name 'Appendix 1'.

## Statistical analysis

Our analyses focused on covering a spectrum of risk and protective factors that have been observed to play a role in injury occurrence among adolescents in school environments. Participants were divided into three groups: those not reporting any injuries throughout the 12 months preceding the survey ('No Injuries'), those reporting one or two injuries ('1–2 Injuries') and those reporting three or more injuries ('3+ Injuries').

The distribution of selected variables within each of the three injury occurrence categories was examined initially. Significant differences between each category and independent variables were explored using Pearson's chi-squared test for categorical variables and multinomial regression for the continuous variable age. Then two multinomial logistic regression (MLR) models were used to examine independent variable associations with those who reported no injuries, those reporting 1–2 injuries and those reporting 3+ injuries while adjusting for covariates. The cut points were selected after distributional aspects (Table 1) in combination with observations in the peer-reviewed literature (*Acquah et al., 2014*). In comparison to binary logistic regression (BLR), MLR allows to examine the probabilities of more than two distinct outcomes. In the study the dependent variable had three possible outcomes, which made MLR more suitable than BLR. The first MLR model included all variables that were shown to be significant in the bivariate analyses ($P < 0.05$). The second model was adjusted only for age and sex. To measure the strength of the associations between the dependent variable and the independent variables we calculated odds ratios (OR) and their respective 95% confidence intervals (CI). All analyses were conducted using the R Statistical Environment for Windows 10 (*R Development Core Team, 2010*).

## RESULTS

Within the recall period (1 year), 30% of sampled adolescents reported having been injured, 24% of whom reported 1–2 injuries and 6% reported three or more injuries. Of the 1–2 Injuries group 57% were male and of the 3+ Injuries group 62% were male. The overall mean age of our sample was 13.4 (SD = 1.7). Those reporting alcohol misuse constituted 11.2%, 36% reported smoking tobacco and 12% reported being involved in a physical fight within the last 12 months. Being a victim of bullying was affirmed by 37.5% while 42.7% reported having bullied others. Of the participating adolescents

**Table 1 Distribution of selected factors according to categories of injury occurrence among school attending adolescents in Greenland (2005/2006).**

|  | No injuries (n = 1,456) | 1–2 Injuries (n = 503) | 3+ Injuries (n = 116) | P-value |
|---|---|---|---|---|
| Age (mean) | 13.4 | 13.4 | 13.2 | 0.480 |
| Gender (male) | 43.4 | 56.7 | 62.1 | <0.001 |
| VPA (≥4 days) | 48.2 | 53.4 | 44.5 | 0.082 |
| VPA outside school (≥4 days) | 49.1 | 52.5 | 60.2 | 0.048 |
| Smoking (Yes) | 35.5 | 35.9 | 39.8 | 0.666 |
| Alcohol misuse (≥2/month) | 10.0 | 12.5 | 14.0 | 0.168 |
| Bullying victimisation (≥2/ month) | 32.6 | 46.9 | 54.4 | <0.001 |
| Bullying perpetrator (≥2/month) | 39.5 | 50.4 | 52.2 | <0.001 |
| Physical fighting (≥2 fights) | 8.7 | 16.4 | 33.3 | <0.001 |
| Job father (skilled job) | 42.4 | 38.4 | 45.3 | 0.352 |
| Job mother (skilled job) | 47.6 | 48.6 | 41.7 | 0.516 |
| Talk to same gender |  |  |  | 0.222 |
| Very easy (n = 701) | 49.3 | 44.0 | 39.8 |  |
| Easy (n = 528) | 34.7 | 38.2 | 38.6 |  |
| Hard (n = 141) | 9.3 | 10.0 | 10.8 |  |
| Very hard (n = 52) | 3.3 | 4.4 | 2.4 |  |
| Does not have/see this person (n = 56) | 3.5 | 3.3 | 8.4 |  |
| Talk to opposite gender |  |  |  | 0.229 |
| Very easy (n = 383) | 28.3 | 28.0 | 29.2 |  |
| Easy (n = 423) | 31.7 | 30.7 | 29.2 |  |
| Hard (n = 281) | 21.6 | 19.8 | 13.9 |  |
| Very hard (n = 135) | 9.1 | 12.5 | 9.7 |  |
| Does not have/see this person (n = 132) | 9.3 | 9.1 | 18.1 |  |
| Self-assessed wealth |  |  |  | 0.478 |
| Well-off (n = 139) | 19.2 | 18.1 | 23.0 |  |
| Average (n = 1,394) | 72.9 | 75.7 | 72.0 |  |
| Not well-off (n = 363) | 7.9 | 6.2 | 5.0 |  |
| Live without mother | 19.9 | 23.6 | 29.4 | 0.024 |
| Live without father | 39.5 | 37.2 | 45.9 | 0.235 |
| Live without both parents | 13.2 | 13.6 | 17.4 | 0.457 |
| Live with both parents | 53.4 | 52.3 | 42.2 | 0.077 |

**Note:**
All variables are expressed as proportions (in %) except for age (mean).

78.7% lived together with their mother. A total of 60.6% of the participating adolescents confirmed living together with their father. Living together with both parents was affirmed by 52.5%. A skilled maternal job was confirmed by 47.1%, while 41.3 % classified their father's job as skilled. When asked to assess their own family's wealth 73.5% assessed it as average, 7.3% as not well-off, and 19.2% as well-off. Practising VPA on 4 or more days
a week was affirmed by 49.7%, while 50.8% practised VPA outside of school 4 days or more a week.

Table 1 depicts the distribution of the independent variables within the three categories of the dependent variable. Significance was found in the psychological and physical aggression category (Bullying victimisation, Bullying perpetrator, Physical fighting) and in social categories such as living together with their mother. Furthermore, being male and practising VPA outside of school settings yielded higher percentages of injury occurrence. Of the aforementioned results only living together with the mother had a protective effect against injury occurrence. All other factors were found to have increased the occurrence of self-reported injuries. No significant differences were found with respect to age, substance use, parental jobs or self-assessed wealth. Further factors that did not prove to be significant were living with fathers, VPA and talking to friends of the same and opposite gender.

After adjusting for all covariates, the first regression model delivered the following results (Table 2). Those who were injured once or twice within the one year preceding the survey were found to been bullied more often (OR = 1.58; CI [1.25–1.99]). Involvement in physical fighting increased the risk for injuries significantly (OR = 1.58; CI [1.14–2.20]). Males were overrepresented among those who reported being injured once or twice (OR = 1.54; CI [1.23–1.93]).

The significant associations between getting injured three times or more were being male (OR = 1.62; CI [1.05–2.51]), being bullied (OR = 2.29; CI [1.48–3.54]) and being involved in a physical fight (OR = 3.42; CI [2.09–5.59]).

The second regression model was adjusted only for age and gender. The results are summarised below in Table 3. Those who reported being in a fight (OR = 1.82; CI [1.33–2.47]), being bullied (OR = 1.81; CI [1.47–2.24]) and having bullied others (OR = 1.53; CI [1.25–1.89]) were more likely to have been injured once or twice. Males were also at a higher risk to be injured once or twice (OR = 1.70; CI [1.39–2.09]).

Significant associations were found between being injured 3 times or more and being male (OR = 2.13; CI [1.44–3.14]), physical fighting (OR = 4.47; CI [2.86–7.01]), being bullied (OR = 2.43; CI [1.65–3.57]) and bullying others (OR = 1.67; CI [1.13–2.45]). Living without a mother was found to be associated with the experience of three or more injuries as well (OR = 1.63; CI [1.05–2.52]).

## DISCUSSION

This study results suggest that the incidence in Greenland for having experienced a medically attended injury at least once within a one-year period of recall was 24.2%, whereas the average cross-national incidence for injuries among school-attending adolescents from all countries that participated in the HBSC survey ranks higher with 34.2%. The Greenlandic incidence for injuries among school-attending adolescents from 9 to 19 years for being injured three times or more was 5.6%. That is also lower than the average cross-national incidence of 9.4% (*Currie et al., 2008*). Most of the participating countries (full list of the participating countries: http://www.hbsc.org/) had a similar or higher GDP and HDI than Greenland at the time the survey was conducted. Greenlandic

**Table 2 Odds ratios and 95% confidence intervals for injury-influencing factors among school attending adolescents in Greenland. Adjusted for all significant variables (2005–2006).**

|  | 1–2 Injuries, OR (95% confidence intervals) | P-value | 3+ Injuries, OR (95% confidence intervals) | P-value |
|---|---|---|---|---|
| Gender | 1.54 [1.23–1.93] | <0.001 | 1.62 [1.05–2.51] | 0.030 |
| VPA outside school | 1.08 [0.86–1.34] | 0.510 | 1.30 [0.85–1.98] | 0.228 |
| Physical fighting | 1.58 [1.14–2.20] | 0.006 | 3.42 [2.09–5.59] | <0.001 |
| Bullying victimisation | 1.58 [1.25–1.99] | <0.001 | 2.29 [1.48–3.54] | <0.001 |
| Bullying perpetrator | 1.26 [1.00–1.59] | 0.051 | 1.15 [0.74–1.79] | 0.530 |
| Live without mother | 1.19 [0.92–1.56] | 0.185 | 1.45 [0.91–2.32] | 0.116 |

**Note:**
Reference category = no injuries.

students also reported lower rates of witnessing other people being injured or killed in comparison to Iceland, the Faeroe Islands or Denmark (*Karsberg, Lasgaard & Elklit, 2012*). Transportation in many areas of Greenland during the summer can only be achieved by air or sea, whilst in the winter travelling on sea ice becomes a valuable traffic component as well (*Grydehøj, 2014*). The lack of large roads and car traffic compared to other highly developed countries may have implications for the occurrence of road traffic related injuries in Greenland. Since those make up around 57% of all unintentional childhood injury deaths, it could be a possible explanation for the observed lower injury incidence rate in Greenland (*Peden et al., 2009*).

Injuries have long been known to be more prevalent amongst males, who accounted for about 68% of all injury-related deaths in 2010 (*Lozano et al., 2012*). It was shown that male adolescents accounted for 57% (1–2 Injuries) and 62% (3+ Injuries) of all injuries in the respective groups.

We did not find evidence for injuries to be associated with alcohol misuse or smoking. Historically, Greenlandic adolescents have been more likely to engage in smoking and binge drinking behaviour (*Niclasen & Bjerregaard, 2007*). Even though the impact of inebriation on injury severity remains unclear, research has found that alcohol consumption and smoking have been linked to the suffering of injuries in adolescents (*Cherpitel et al., 2015*; *Knapik & Bedno, 2018*; *Peden et al., 2009*; *Valdez et al., 2016*). The mean age of our study's participants was 13.4 years (SD = 1.74). It was pointed out that alcohol-attributable deaths among adolescents (0–15 years of age) only account for 0.1 percent of all adolescent deaths worldwide (*Rehm & Shield, 2013*). *Skala & Walter (2013)* also considered the age of 15 to be a threshold after which a rise in repetitive excessive alcohol consumption can be observed. Further, the HBSC international report from 2005/06 points out significantly higher rates of drunkenness among 15 year olds than among 11 or 13 year olds (*Currie et al., 2008*). This potentially illustrates that the present study sample may have been too young to be exposed to excessive alcohol consumption behaviour, hence significant associations could not be found. Lastly, it remains difficult to assess alcohol consumption through a questionnaire as the imprecise recall of alcohol consumption may drive an association to the null (*Wilson et al., 2012*). The prevalence of smoking in the present sample was found to be higher than in other high-income

**Table 3 Odds ratios and 95% confidence intervals for injury-influencing factors among school attending adolescents in Greenland. Adjusted for age and gender (2005/2006).**

| | 1–2 Injuries, OR (95% confidence intervals) | P-value | 3+ Injuries, OR (95% confidence intervals) | P-value |
|---|---|---|---|---|
| Gender | 1.70 [1.39-2.09] | <0.001 | 2.13 [1.44-3.14] | <0.001 |
| VPA outside school | 1.05 [0.85-1.30] | 0.652 | 1.38 [0.93-2.06] | 0.108 |
| Physical fighting | 1.82 [1.33-2.47] | <0.001 | 4.47 [2.86-7.01] | <0.001 |
| Bullying victimisation | 1.81 [1.47-2.24] | <0.001 | 2.43 [1.65-3.57] | <0.001 |
| Bullying perpetrator | 1.53 [1.25-1.89] | <0.001 | 1.67 [1.13-2.45] | <0.001 |
| Live without mother | 1.21 [0.94-1.56] | 0.133 | 1.63 [1.05-2.52] | 0.028 |

**Note:**
Reference category = no injuries.

countries. Almost 26% of all participants reported smoking daily and 36% report smoking more or less regularly. In comparison, daily smoking prevalence for US American high school students ranks between 9% and 22% (*Siqueira et al., 2000*). The association between smoking and injury has been subset into four categories of reasoning: direct toxicity, distractibility, associated medical conditions, and confounding factors (*Sacks & Nelson, 1994*). Since the theories of direct toxicity and distractibility are either still questionable or only applicable to drivers, they do not aid to explain a possible relationship between injuries and smoking within our study's cohort. Medical conditions associated with the toxic effects of smoking such as cardiovascular diseases or cancer, usually do not appear in teenagers but rather during adulthood (*Burns, 2003*). The most frequent injuries resulting from smoking are burns, but burns only make up about 3.3% of all unintentional childhood injury deaths in high-income countries (*Peden et al., 2009*; *Sacks & Nelson, 1994*). The missing applicability of the study conducted by Sacks and Nelson as well as the rather small share of burns in injury deaths could explain why the present study is silent on an association between smoking and injuries.

Consistent with existing research, bullying victimisation as well as bullying others was associated with increased injury occurrence. Being involved in bullying as a victim or perpetrator goes along with an increased risk of physical fighting (*Rudatsikira, Muula & Siziya, 2008*) which can lead to injuries requiring medical attention. Physical fighting was, as mentioned, a significant risk factor for suffering from injury in the present study.

The finding that living without mother increases risk for injury is supported by evidence which shows that an increase of adults living within a household goes along with a decrease in the risk of injury (*Haynes, Reading & Gale, 2003*). Parents also play a deciding role in teaching their children about injury prevention, thus the reduction of parental contact may put adolescents at risk for injuries (*Wong & Breslin, 2017*). Moreover, a study examining parenting differences between mothers and fathers came to the conclusion that especially uninvolved maternal parenting can lead to delinquency, depression, and decreased school commitment (*Simons & Conger, 2007*). These findings may aid to demonstrate the protective effect involved parenting can have on the risk of adolescent injury occurrence. Our results only partially support these claims since we could show that

living without a mother was associated with an increase in injury occurrence whereas living without a father showed no association with injury occurrence.

This study provides valuable insight into a largely neglected area of global health research. To our knowledge this is the only study to have examined the epidemiology of injuries among a school-based sample in Greenland. Despite that, the current study also had the following limitations, since, besides giving valuable insight, it also remains confined and silent on several topics.

First, the study does only include students that go to school and it does not include adolescents that do not attend school at all. Research shows that those who are not in school may be at even greater risk for injuries, yet assessing this would go beyond the scope of this study (*Dingli et al., 2012*).

Secondly, some questions such as drug use, bullying, or physical fighting might have altered our findings in in the form recall bias, as the participants had to recall information within a rather large time frame which might have reduced their recall ability and accuracy (*Celedonia et al., 2013*; *Wilson et al., 2012*). Also, all the questions were self-reported, which may have been subject to social desirability and non-response bias (*Acquah, Wilson & Doku, 2014*).

Lastly, the cross-sectional nature of the dataset prevents us from drawing causal inference from our findings. The results must be interpreted within their own time context. Longitudinal data is necessary to establish the aforementioned causal pathways and to take into account the nature of change over time. Still our results have value not only for the historical aggregation of data, but they also play a role in understanding how young populations, and their behaviours, might change over time.

The body of research on injuries in circumpolar regions, especially Greenland, is rather limited at present. Therefore, this study also stresses the need for further examination of this topic.

## CONCLUSIONS

The results of this study suggest that interpersonal relationships and conflicts are heavily linked with injury occurrence among school-attending adolescents in Greenland. Future programmes should consider the risk behaviours that were found to have a negative association with the adolescents' well-being such as bullying and physical fighting. Moreover, further research would benefit from exploring the kinds of injuries Greenlandic adolescents suffer to identify factors that might modify the associations pointed out in this study.

### Funding

We received financial support from the Else Kröner-Fresenius-Stiftung within the Heidelberg Graduate School of Global Health, by the Baden-Württemberg Ministry of Science, Research and the Arts and by Ruprecht-Karls-Universität Heidelberg. The funders

had no role in study design, data collection and analysis, decision to publish, or
preparation of the manuscript.

## Grant Disclosures
The following grant information was disclosed by the authors:
Else Kröner-Fresenius-Stiftung within the Heidelberg Graduate School of Global Health,
by the Baden-Württemberg Ministry of Science, Research and the Arts and by Ruprecht-
Karls-Universität Heidelberg.

## Competing Interests
The authors declare that they have no competing interests.

## Author Contributions
- Louis Jansen conceived and designed the experiments, performed the experiments,
  analysed the data, prepared figures and/or tables, authored or reviewed drafts of the
  paper, and approved the final draft.
- Till Bärnighausen conceived and designed the experiments, analysed the data, authored
  or reviewed drafts of the paper, and approved the final draft.
- Michael Lowery Wilson conceived and designed the experiments, analysed the data,
  authored or reviewed drafts of the paper, and approved the final draft.

## Data Availability
The raw data are available in the Supplemental Files.

## Supplemental Information
Supplemental information for this article can be found online at http://dx.doi.org/10.7717/
peerj.8605#supplemental-information.

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
