# Peer review of "Injuries among adolescents in Greenland: behavioural and socio-economic correlates among a nationally representative sample"

_PeerJ, doi:10.7717/peerj.8605_

## Round 0.1 · original submission · Major Revisions

Dear authors,

After asessing the reviewers' reports, I think your manuscript has scientific merit to be published in PeerJ. However, there are some issues which you must address in a revised version of the text.

Best regards,
Dr Palazón-Bru (academic editor for PeerJ)

·

Basic reporting

The submission is written in clear unambiguous and technically correct English. However, the intro and background can be improved, as well as the structure of the text.

Experimental design

The research is conducted in conformity of the ethical standers, but the experimental design can be improved: Research questions and/or hypothesis and relevance of the study are missing, and methods should be described with sufficient information, since reproduction is not possible with the information provided.

Validity of the findings

The data are robust and statistically sound, but they are not up to date. That is not a necessity for all studies, for example to investigate theoretically based research questions, but for a study investigating the prevalence of phenomena and their determinants it is necessary. Speculations and conclusions can be improved by linking them to research questions and results of the study.

Additional comments

See the attachment

·

Basic reporting

I find the language fully acceptable – but I am not a native English speaker. The structure seems to confirm to the standard of the journal.

The tables are relevant. the data are robust, the statistics are excellent, the subject is very relevant and is a useful contribution to the knowledge about injuries in Greenlandic adolescents – which as stated is very sparse.

Experimental design

Fully acceptable

Validity of the findings

Se below

Additional comments

I find the language fully acceptable – but I am not a native English speaker. The structure seems to confirm to the standard of the journal.

The tables are relevant. the data are robust, the statistics are excellent, the subject is very relevant and is a useful contribution to the knowledge about injuries in Greenlandic adolescents – which as stated is very sparse.

Despite this, the paper needs a major revision due to serious limitations.

The study has a serious limitation: The authors have a total lack of knowledge about the context the data has been gathered in.

They also seems to assume that the ways of living are the same all over the Arctic region and therefor mainly use US or Canadian literature on areas where Greenlandic literature can be found rather easy.

This lack of local knowledge have the result that most of the discussion is pure speculations For some of the speculations the authors draw conclusions that are not only wrong, but unacceptable wrong.

Specific comments
Line 59 - 61. Population data should be updated
Line 68-69 Data on suicides should be updates, use examples data from the office og the Chief Medical Officer or Statistics Greenland.
Line 86 Either exclude Greenland in the sentence or include a Greenlandic reference.
Line 92 It I correct that no other publications in English exist, still grey literature such as the national reports from the HBSC study are published.
Line 143: Is the selection of cut-points according to the literature or how were they selected?
Line 191. The International data from the HBSC study only include the agegroups 11, 13, 15.years.
Line 193 “thrice” – is the word really used in ordinary language?
Line 200 There are healthcare providers in all Greenlandic towns and villages, although not doctors in some smaller towns and in villages.
Line 202-205 This is pure speculation – can the authors find any documentation. Half of the year it is daylight 24/7 – doesn’t that count as well? May be the lack of road accidents is due to lack of cars in many villages!
Line 213-214 .. despite that Greenlandic adolescents are binge drinking – as in the other Nordic countries.
Line 220-223 The speculation could be checked by consulting the international HBSC report from the 2005/2006 study.
Line 241 Please reality check the statement about the suicide rate as a cause of decreased fatherhood involvement in Greenlandic fathers – it is not correct.
Line 243 There are numerous Greenlandic articles about suicide why not use a Greenlandic reference.
Line 250-261 this is completely wrong. Canadian ways to look at education and at the school system cannot be applied to Greenland. And the words are not in Greenlandic but in the language used by Canadian Inuits. The Greenlandic school system and ways of thinking about education is very highly influenced by Danish and Nordic traditions which and is very different from the Canadian ways.
Line 261-264 wrong again. Please correct.

---

## Round 0.2 · Major Revisions

Dear authors,

The revision is still pending some changes which you should address in a revised version of the text.

Best regards,
Dr Palazón-Bru (academic editor for PeerJ)

·

Basic reporting

See pdf and below

Experimental design

see pdf and below

Validity of the findings

see pdf and below

Additional comments

Review of the revised manuscript “Injuries among adolescents in Greenland: behavioural and socio-economic correlates among a nationally representative sample”

I thank the authors for their rebuttal letter, which is very clear. I also thank them for the revision of their manuscript on injuries among adolescents. I must say the revision is a serious improvement. Congratulations for that. I have some minor comments, some recommendations and some questions.

Introduction
The authors provide a clear purpose of the study: investigating the socioeconomic and behavioural correlates associated with injury occurrence. However, I wish to provide three recommendations.
1. A difference between intentional and unintentional injuries are provided, but it is not mentioned whether both kinds of injury are under investigation, or only one of them. This is important for understanding the discussion.
2. In the abstract it is mentioned that the primary outcome is measuring the experience of injury. This does not match with the newly formulated purpose in the introduction.
3. The introduction is on violence and injury. At the end of the introduction they refer to Schnohr and Niclasen who found an increase on bullying. This term is not mentioned earlier. Although bullying can be perceived as a behavioural correlate associated with injury, the authors should provide a link to injury and violence on the one hand, and bullying on the other.

Materials and methods
I understand the argument of the authors concerning the time limitation of the data. And I agree with them, concerning the purpose of the research. Most comments on the measurement part are not relevant anymore. Here I have four recommendations and one question.
1. Although there is a link to the website for more information, it is convenient to provide some information on the data collection methods, for example the situation in which the questionnaire was completed.
2. The cut points for the independent variable is mentioned twice: under the measurements as well as under the statistical analysis. Since in the latter part the base for this choice for these cut points is provided, they can be deleted in the measurement part.
3. The first part of the statistical analysis concerns the measurements. This part should be moved the description of the measurements.
4. As is mentioned in the comments in the rebuttal letter, injury is the dependent variable and all the others are the independent variables. However, ‘Significant differences between each category and independent variables were explored using … ANOVA for the continuous variable age”. I don’t understand how the ANOVA is conducted. This sentence suggests that in this analysis a continuous variable (age) is the independent variable and a categorical variable (injury) is the dependent one. In ANOVA the independent variable should be the categorical and the dependent should be the continuous one.
5. Since for the measurement of the independent variables the authors refer to a website, and no information is provided about the answering categories of these variables, it is more convenient to mention there that these variables are categorical and to provide some examples.

Results
Two recommendations on the results:
1. Some more frequencies are reported, but they are not completely clear. It is mentioned that parent socio-economic status and self-assessed wealth are independent variables, but frequencies are provided for mothers’ SES, for fathers’ SES and for families’ SES. And I don’t understand the frequencies for ‘living with parents’: of the participating adolescents, 78% lived with their mother and 60% lived with their father. Together that’s 138%. Also, frequencies for physical activity (VPA) and VPA (outside school hours) are missing.
2. Table 1 is very clear. Except for the living circumstances. An extra line for adolescent who live with both parents makes it complete.

Discussion
The discussion is seriously improved. I have one question and one suggestion.
1. As mentioned above, it is not clear whether the authors measured intendent or unintended injuries, or both. For the lower prevalence, they provide the explanation concerning transportation in Canada. Since my problem concerning the definition of HBSC is cleared in this revision, I don’t understand why the authors deleted the explanation concerning difficulties seeing a healthcare provider (and the explanation concerning time spending indoors).
There is a lot of literature on parenting style and difference between mothers and fathers. These studies mostly suggest that mothers are more protective than fathers. That may be an explanation for the finding that living without a mother increases injuries, whereas living without a father does not.

·

Basic reporting

Regarding "Injuries among adolescents in Greenland: behavioural and socio-economic correlates among a nationally representative sample" is changes to be much more basis on Greenlandic realities and research and not NorthAmerican making it much more reliable and interesting.

There are still som minor changes needed:

Line 213 ff: The sentences about inclusion I not correct as is can be read at the international homepage. All schools in Greenland is invited to participate. The parents do not give informed consent. The school board decide if the school want to participate.

Comments to the result section: In Greenland two peoples and to major languages is used a relevant item to look at I violence in different population groups, either using “Where is your father/mother born”, perceived ethnicity or language performance.

After these revisssions it is my view that it can be published and that it actually can contribute to the sparse research on the topic in Greenlandic adolescents.

Experimental design

See above

Validity of the findings

See above

Additional comments

Se above

---

## Round 0.3 · accepted · Accept

All the points raised by the reviewers have been correctly addressed.

There are some final points by R1 which you should address while in production.

·

Basic reporting

see below

Experimental design

see below

Validity of the findings

see below

Additional comments

I wish to thank the authors for the rebuttal letter and the improvements of their paper.

In the second review I wrote that the first part of the statistical analysis concerns the measurements, and that this part should be moved the description of the measurements. The authors reply with the remark that it is changed. The statistical analyses descriptition however still contains remarks about the measurement, which have to be part of the measurement ( ‘The question ‘During the past 12 months, how many times have you been injured?’ had five response options……. While both intentional and unintentional injuries were included’.)

In the second review I also wrote that percentages in the results section are not clear: of the participating adolescents, 78% lived with their mother and 60% lived
with their father. Together that’s 138%. The authors mention that they are not to be understood as cumulative percentages. But then they have to reformulate the text, since now it seems like a typo.

In the second review I asked for an extra line in Table 1 for adolescents who live with both parents. This extra line would make it complete. The authors indeed placed this extra line, but now the percentages for no injuries are not correct: 19.9 is living without mother, 39.5 is living without father, 13.2% is living without both parents and 53.4% is living with both parents. This counts up to 126%. The same for 1-2 injuries and 3+ injuries. So somewhere the percentages are wrong, or the categories are wrong, because an overlap in these categories is not possible.

It have noticed that the concepts in the text are different from the concepts in the table. For example, in the text it is called ‘maternal socioeconomic status and in the table it is called ‘’ job mother (skilled job)’. The same for father’s SES and family wealth with below and above average (in the table self-assessed wealth with well-off, average and not well-off).